# Hydroxyurea does not reverse functional alterations of the nitric oxide-cGMP pathway associated with priapism phenotype in corpus cavernosum from sickle cell mouse

Dalila Andrade Pereira[1], Danillo Andrade Pereira[1], Pamela da Silva Pereira[2], Tammyris Helena Rebecchi Silveira[1], Fabiano Beraldi Calmasini[3], Leonardo Oliveira Reis[4], Fernando Ferreira Costa[2], Fábio Henrique Silva[1] *

1 Laboratory of Pharmacology, São Francisco University Medical School, Bragança Paulista, São Paulo, Brazil, 2 Hematology and Hemotherapy Center, University of Campinas, Campinas, São Paulo, Brazil, 3 Escola Paulista de Medicina, Department of Pharmacology, Universidade Federal de São Paulo, São Paulo, São Paulo, Brazil, 4 UroScience, Pontifical Catholic University of Campinas, Campinas, São Paulo, Brazil

☯ These authors contributed equally to this work.
* fabiohsilva87@gmail.com

## Abstract

Sickle cell disease (SCD) is a genetic disorder that has been associated with priapism. The role of hydroxyurea, a common SCD therapy, in influencing the nitric oxide (NO)-cGMP pathway and its effect on priapism is unclear. To investigate the effect of hydroxyurea treatment on smooth muscle relaxation of corpus cavernosum induced by stimulation of the NO-cGMP pathway in SCD transgenic mice and endothelial NO synthase gene-deficient (eNOS$^{-/-}$) mice, which are used as model of priapism associated with the low bioavailability of endothelial NO. Four-month-old wild-type (WT, C57BL/6), SCD transgenic, and eNOS$^{-/-}$ male mice were treated with hydroxyurea (100 mg/Kg/day) or its vehicle (saline) daily for three weeks via intraperitoneal injections. Concentration-response curves for acetylcholine (ACh), sodium nitroprusside (SNP), and electrical field stimulation (EFS) were generated using strips of mice corpus cavernosum. The SCD mice demonstrated an amplified CC relaxation response triggered by ACh, EFS, and SNP. The corpus cavernosum relaxation responses to SNP and EFS were found to be heightened in the eNOS$^{-/-}$ group. However, the hydroxyurea treatment did not alter these escalated relaxation responses to ACh, EFS, and SNP in the corpus cavernosum of the SCD group, nor the relaxation responses to EFS and SNP in the eNOS$^{-/-}$ group. In conclusion, hydroxyurea is not effective in treating priapism associated with SCD. It is likely that excess plasma hemoglobin and reactive oxygen species, which are reported in SCD, are reacting with NO before it binds to GCs in the smooth muscle of the corpus cavernosum, thus preventing the restoration of baseline NO/cGMP levels. Furthermore, the downregulation of eNOS in the penis may impair the pharmacological action of hydroxyurea at the endothelial level in SCD mice. This study emphasize the urgency for exploring alternative therapeutic avenues for priapism in SCD that are not hindered by high plasma hemoglobin and ROS levels.

**Data Availability Statement:** All relevant data are within the manuscript and its Supporting information files.

**Funding:** Fábio H. Silva and Fernando F. Costa thank São Paulo Research Foundation (FAPESP) for financial support (2019/18886-1, 2017/08122-9). The funders had no role in study design, data collection and analysis, decision to publish, or preparation of the manuscript.

**Competing interests:** The authors have declared that no competing interests exist.

## Introduction

SCD is a genetic blood disorder caused by the production of an abnormal hemoglobin, called HbS. Approximately 100,000 Americans have SCD, this is a disease of global incidence that affects millions of people, but it is more common in the African continent [1, 2]. The primary phenomenon of SCD involves the polymerization of deoxygenated HbS, which leads to the rigidity of red blood cells and their transformation into a sickle shape. This results in hemolytic anemia, vasoocclusion, and subsequent damage to organs [3]. Patients with SCD have several changes such as stroke, leg ulcer, acute chest syndrome, pulmonary hypertension, voiding dysfunction and priapism [3]. Priapism is defined as a prolonged and often painful penile erection, triggered or not by sexual stimulation. The events of prolonged erections cause changes in penile tissue, such as fibrosis and necrosis that can progress to permanent erectile dysfunction [4]. About 30–45% of male patients with SCD have priapism [4, 5]. In patients with SCD, the occurrence rate of erectile dysfunction, specifically linked with recurrent ischemic priapism, is reported to be a minimum of 47.5% [5].

Experimental studies have reported that the main change that occurs in the penis for the development of priapism in SCD is the reduction of NO bioavailability under basal conditions in the penis [6–13]. The penis of men and mice with SCD exhibit lower expression of the endothelial nitric oxide synthase (eNOS) enzyme and increased production of superoxide anion, which reacts with NO, thus reducing the bioavailability of NO [7, 9, 14–16]. Less stimulation of sGC by basal NO leads to reduced levels of cGMP within the smooth muscle cell of corpus cavernosum [10, 11, 17]. cGMP regulates the expression of PDE5 [18]. Due to the reduced levels of cGMP, the penis of mice and men with SCD shows lower activity and expression of PDE5 in the penis [6, 7, 9, 10]. Therefore, when an erectile stimulus occurs, NO activates GCs that convert GTP to cGMP, but cGMP is not efficiently degraded by reduced PDE5 activity, so high levels of cGMP accumulate in the smooth muscle cell of the corpus cavernosum, promoting smooth muscle relaxation and finally, prolonged penile erection (priapism) [6, 7, 9, 10]. Endothelial NO synthase gene-deficient (eNOS$^{-/-}$) mice exhibit exacerbated relaxation of corpus cavernosum associated with reduced PDE5 expression in the penis, indicating that reduced NO bioavailability generates a priapism phenotype [9, 11, 14].

Hydroxyurea, known to inhibit ribonucleotide reductase, was the first drug therapy approved by the U.S. Food and Drug Administration to treat patients with sickle cell anemia [19]. Hydroxyurea is used for the treatment of myeloproliferative syndromes, and in SCD it has beneficial therapeutic effects by acting mainly through increasing levels of fetal hemoglobin (HbF), that acts to impede the polymerization of deoxygenated HbS [20]. Hydroxyurea induces gamma-globin gene expression that will result in the production of HbF through a dependent mechanism of the NO-sGC-cGMP signaling pathway [21]. Previous studies have reported that hydroxyurea can be oxidized by heme groups to generate NO in vivo in animals and patients with SCD [22, 23]. The administration of hydroxyurea increases the levels of NO metabolites (nitrite and nitrate) and cGMP in the plasma of patients with SCD [24].

Clinical studies show that hydroxyurea therapy increases the life expectancy of patients with SCD [19, 25]. However, few clinical studies have reported the beneficial effect of hydroxyurea therapy on priapism in men with SCD (1–3). To date, no studies have investigated the effect of hydroxyurea on priapism or erectile function in an experimental model. Since priapism in SCD is associated with low bioavailability of endothelial NO in the penis, and hydroxyurea acts by increasing NO levels, we hypothesized that hydroxyurea treatment may have beneficial effects on CC dysfunctions associated with priapism. The objective of this research was to examine the impact of hydroxyurea treatment on the relaxation of corpus cavernosum

smooth muscle, brought about by stimulation of the NO-cGMP pathway, in both SCD transgenic mice and eNOS$^{-/-}$ mice.

## Materials and methods

### Ethics statement

This study's experimental protocols were conducted in compliance with the ethical standards for animal use set forth by the Brazilian Society of Laboratory Animal Science (SBCAL) and the EC Directive 86/609/EEC for Animal Experiments. The protocols received approval from the University of Campinas' institutional Committee for Ethics in Animal Experimentation (IACUC/CEEA-UNICAMP, Permit number 5729-1/2021). For the experiments, mice were anesthetized using 100 mg/kg Ketamine and 10 mg/kg Xylazine administered through an intraperitoneal injection, with utmost care taken to mitigate any distress to the animals.

### Animals and treatment

Male mice, aged four months, belonging to six groups—wild type (WT, C57BL/6), SCD transgenic, and eNOS$^{-/-}$, were exposed to a treatment regimen of hydroxyurea (100 mg/Kg/day) or its vehicle (saline) for a duration of three weeks. This treatment was administered via intraperitoneal injections [26]. Sourced from Jackson Laboratories (Bar Harbor, ME), the mice were subsequently produced and studied at the Multidisciplinary Center for the Investigation of Biological Science in Laboratory Animals, University of Campinas. The sickle cell transgenic mouse model, which is homozygous, was created using a "knock-in" approach. This involved the replacement of mouse α-globin genes with corresponding human α-globin genes, while simultaneously substituting mouse β-globin genes with human Aγ and βS (sickle) globin genes [27].

### Functional studies in corpus cavernosum strips and concentration-response curves

Corpus cavernosum samples were collected from mice under the effects of ketamine and xylazine anesthesia. These samples were then installed in a 7-mL organ system filled with Krebs-Henseleit solution maintained at 37°C and continuously aerated with a 95% O2 and 5% CO2 mixture (pH 7.4). Isometric force alterations were registered using a strip myograph designed for isometric force documentation (model 610M; Danish Myo Technology, Aarhus, Denmark), working in conjunction with a data acquisition system (PowerLab 8/30, LabChart 7; ADInstruments, Sydney-NSW, Australia). Initial resting tension was set to 2.5 mN when the experiments began. A preparatory period of 60 minutes was established, during which the bathing medium was refreshed every 15 minutes leading up to the experiments. Cumulative concentration-response curves were then developed for both the muscarinic agonist, acetylcholine (ACh; 1 nM—10 μM), and the nitric oxide-donor compound, sodium nitroprusside (SNP; 10 nM—100 μM) in tissue strips that were precontracted with phenylephrine (3–10 μM).

### Electrical-field stimulation (EFS) in corpus cavernosum strips

The cavernosal strips were positioned between two platinum electrodes linked to a Grass S88 stimulator (Astro-Med Industrial Park, RI, USA) for EFS. The parameters set for EFS included a voltage of 50 V, pulse width of 1 ms, and 10-second trains of stimuli at varying frequencies. All preparations were studied under maximum voltage to determine frequency-response relationships during electrical stimulation. In preparation for the study of nitrergic cavernosal

relaxations, tissues were pretreated with guanethidine (30 μM; for the depletion of adrenergic fiber catecholamine stores) and atropine (1 μM, to induce muscarinic receptor antagonism) before being pre-contracted with phenylephrine (3–10 μM). Once a stable contraction level was reached, a sequence of EFS-induced relaxations were formed (2–32 Hz) [28]. The data collected were then normalized relative to the maximum changes from the phenylephrine-induced contraction in each tissue, which was set as the 100% mark.

## Statistical analysis

Statistical analysis was conducted utilizing the GraphPad Prism program (GraphPad Software Inc., San Diego, CA, USA). The results are presented as the mean ± S.E.M. derived from N experiments. Statistical differences were identified through a one-way analysis of variance (ANOVA), followed by a post-test using the Tukey method. A value of $P < 0.05$ was considered statistically significant.

## Results

### Effect of hydroxyurea treatment on acetylcholine-induced relaxation in corpus cavernosum of SCD mice

Endothelium-dependent relaxation was evaluated by concentration-response curves to acetylcholine (ACh, 1 nM—10 μM) in corpus cavernosum of mice pre-contracted with phenylephrine (Fig 1A–1C). Values of ACh maximal response (Emax) were significantly higher ($P < 0.05$) in the corpus cavernosum of the SCD+V mice compared to the WT+V mice (Fig 1A and 1D). Treatment with hydroxyurea did not change the ACh Emax in the corpus cavernosum of the SCD+HU (Fig 1B) and WT+HU groups (Fig 1C and 1D).

### Effect of hydroxyurea treatment on sodium nitroprusside-induced relaxation in corpus cavernosum of SCD and eNOS$^{-/-}$ mice

Endothelium-independent relaxation was evaluated by concentration-response curves to SNP (1 nM—100 μM) in corpus cavernosum of mice pre-contracted with phenylephrine (Fig 2A–2E). Values of SNP maximal response (Emax; Fig 2F) were significantly higher ($P < 0.05$) in the corpus cavernosum of SCD and eNOS$^{-/-}$ mice compared to WT+V mice (Fig 2A and 2D, respectively). Treatment with hydroxyurea did not modify the values of SNP Emax (Fig 2F) in the corpus cavernosum of the SCD+HU, WT+HU and eNOS$^{-/-}$+HU groups (Fig 2B, 2C and 2E, respectively).

### Effect of hydroxyurea treatment on EFS-induced relaxation in corpus cavernosum of SCD and eNOS$^{-/-}$ mice

EFS of the preparation induced frequency-dependent relaxations (2–32 Hz) in corpus cavernosum of mice pre-contracted with phenylephrine (Fig 3). Nitrergic relaxations were significantly greater in the corpus cavernosum of the SCD-V and eNOS$^{-/-}$-V groups compared to the WT+V at all frequencies studied (Fig 3). Treatment with hydroxyurea did not modify nitrergic relaxation in the WT+HU, SCD+HU and eNOS$^{-/-}$+HU (Fig 3).

## Discussion

Penile erection is achieved after relaxation of the smooth muscle of the corpus cavernosum and penile arteries. This process is initiated by NO released from nitrergic fiber, which is produced by nNOS from L-arginine. It is currently accepted that erection is maintained by

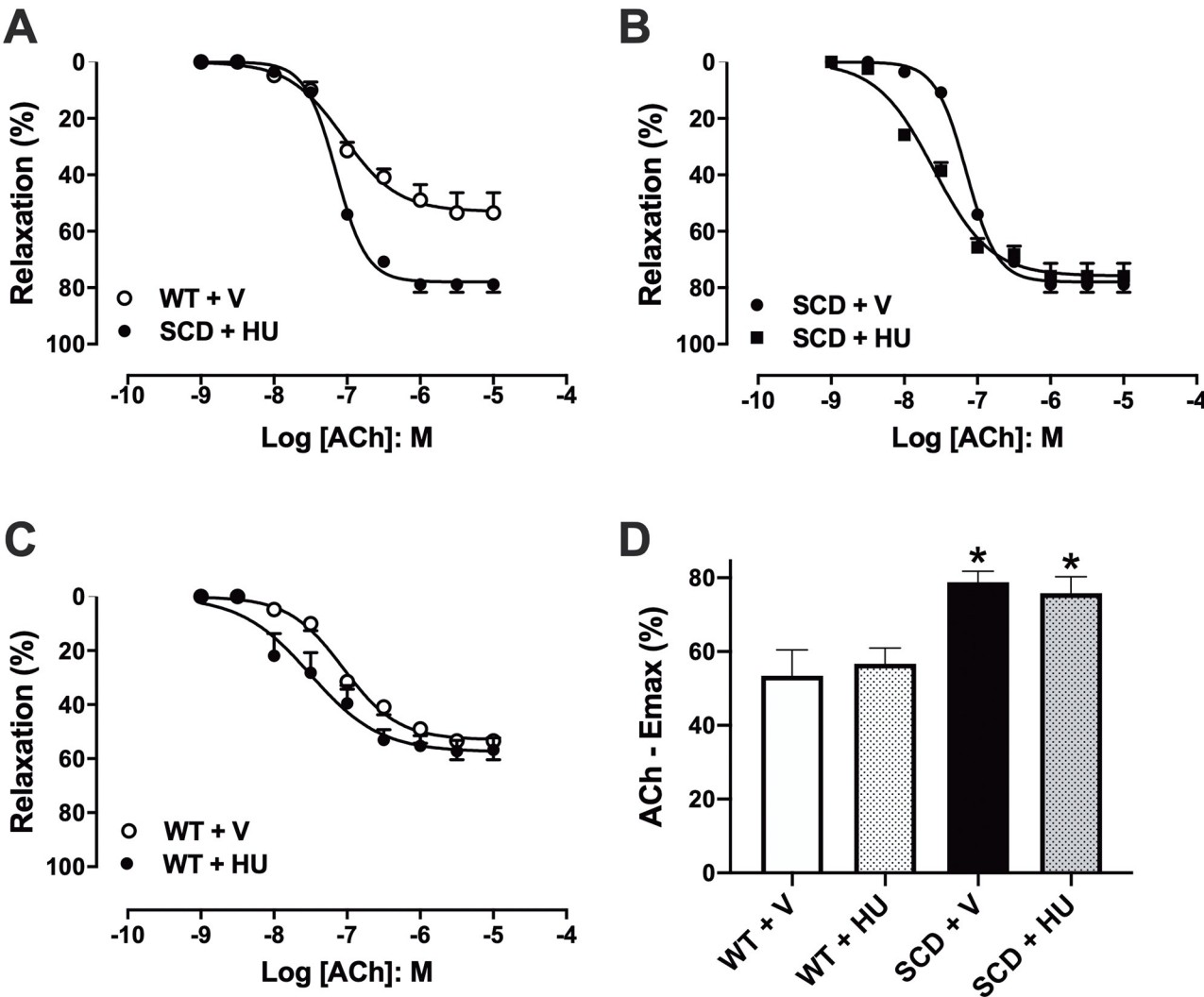

**Fig 1.** (A-C) Concentration-response curves to acetylcholine (ACh) in corpus cavernosum of WT and SCD mice treated with vehicle or hydroxyurea (100 mg/kg/day, for 3 weeks, via IP). (D) Maximum response values (Emax) for ACh. * P < 0.05 compared to the WT-V group. V, vehicle. HU, hydroxyurea. Data represent the mean ± SEM for 5 mice in each group.

endothelial NO produced by eNOS in association with NO generated by nNOS [29]. After being produced, NO diffuses to the smooth muscles of the corpus cavernosum, which in turn activates GCs, generating cGMP, and finally, relaxation of the corpus cavernosum and penile erection [30]. Relaxation of the smooth muscle of the corpus cavernosum and penile erection is terminated by the metabolism of cGMP by the enzyme PDE5 [30]. Therefore, we evaluated EFS-induced corpus cavernosum relaxation (nitrergic relaxation) and ACh-induced endothelium-dependent relaxation. The SNP is an exogenous NO-donor compound, widely used to assess endothelium-independent relaxation. In our study, corpus cavernosum relaxation induced by EFS and SNP was greater in the eNOS$^{-/-}$ and SCD group, as well as the relaxation induced by ACh in the SCD group. These results agree with previous studies that showed that increased corpus cavernosum relaxation after stimulation of the NO-cGMP pathway (ACh, EFS, SNP) is associated with decreased PDE5 expression/activity in the erectile tissue of SCD mice. and eNOS$^{-/-}$ [11, 17, 31].

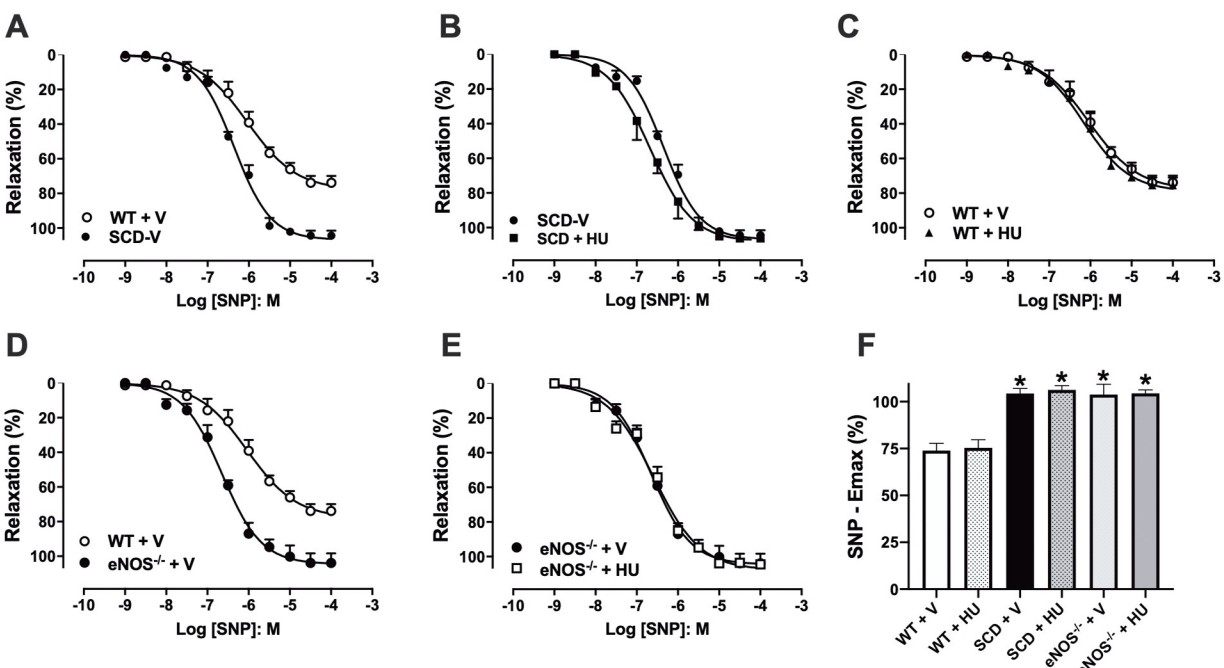

**Fig 2.** (A-E) Concentration-response curves to sodium nitroprusside (SNP) in corpus cavernosum of WT, SCD and eNOS$^{-/-}$ mice treated with vehicle or hydroxyurea (100 mg/kg/day, for 3 weeks, via IP). (F) Maximum response values (Emax) for SNP. * P < 0.05 compared to the WT-V group. V, vehicle. HU, hydroxyurea. Data represent the mean ± SEM for 5 mice in each group.

PDE5 expression is upregulated by cGMP levels [18]. In SCD mice, decreased expression of eNOS results in reduced bioavailability of NO in the penis [10, 17]. Furthermore, the increased production of reactive oxygen species (ROS) also contributes to the reduction of NO bioavailability by interacting with NO in the penis of mice and men with SCD [9, 10, 15]. It is currently

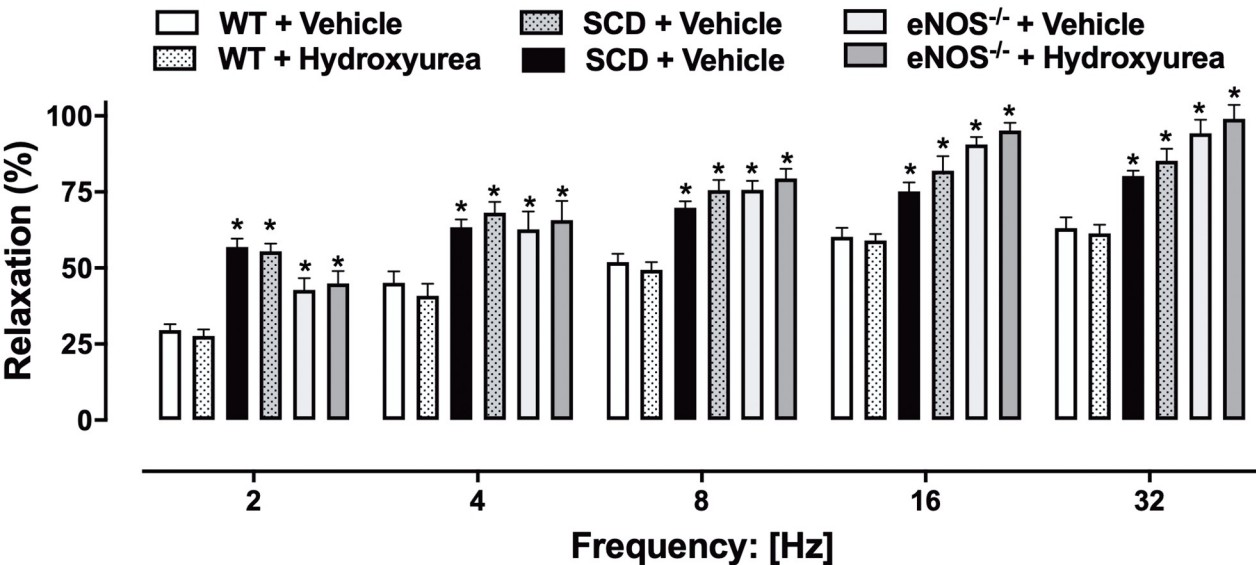

**Fig 3. Nitrergic relaxation induced by EFS in corpus cavernosum of WT, SCD and eNOS$^{-/-}$ mice treated with vehicle or hydroxyurea (100 mg/kg/day, for 3 weeks, via IP).** * P < 0.05 compared to the WT-V group. V, vehicle. HU, hydroxyurea. Data represent the mean ± SEM for 5 mice in each group.

accepted that reduced NO/cGMP bioavailability is a fundamental alteration for the development of priapism in SCD, which leads to a lower expression of PDE5 in erectile tissue [32].

A pharmacological strategy to normalize PDE5 expression and relaxation of the corpus cavernosum is by increasing the basal bioavailability of NO/cGMP in the penis. [6, 7, 9, 10, 15]. Previous studies have reported that hydroxyurea increases the bioavailability of NO and cGMP in patients with SCD [22, 23]. Therefore, in the present study, we initially investigated the treatment with hydroxyurea in SCD mice. Treatment with hydroxyurea for 3 weeks did not modify the exaggerated corpus cavernosum relaxations induced by ACh, EFS and SNP in the SCD group.

A previous study reported that hydroxyurea increases NO production via phosphorylation of eNOS at Ser1177 in a PKA-dependent manner in endothelial cells [21]. The non-selective NOS inhibitor, L-NAME, abolished the effect of hydroxyurea in stimulating NO production, indicating that this effect is an eNOS-dependent mechanism. It is well established that the penises of mice and men with SCD display lower protein expression of eNOS (8,10–12). It is likely that the lower expression of eNOS in the corpus cavernosum impairs the effect of hydroxyurea in increasing the bioavailability of NO in SCD. Furthermore, excess plasma hemoglobin and increased production of superoxide anion in the penis, which are reported in SCD mice [8, 11, 16, 33], may be reacting with the NO donated by hydroxyurea before it reaches GCs in the smooth muscle cells of the corpus cavernosum. In rats, the metabolism of hydroxyurea to generate NO occurs mainly in the liver [22]. Treatment with hydroxyurea increased eNOS protein expression in mouse kidney vessels [34].

A recent study showed that a new hybrid compound derived from resveratrol with nitric oxide donor property (RVT-FxMe) did not modify the phenotype of priapism in SCD mice due to the excess of hemoglobin in the plasma that inactivated the NO donated by the compound [11]. On the other hand, in eNOS$^{-/-}$ mice that do not show elevated plasma hemoglobin levels, treatment with the RVT-FxMe compound reversed the priapism phenotype associated with increased relaxation of corpus cavernosum induced by stimulation of the pathway NO-cGMP [11]. In our study, to identify whether eNOS deficiency impairs the action of hydroxyurea in the corpus cavernosum, we treated eNOS$^{-/-}$ mice for 3 weeks. Treatment with hydroxyurea also did not modify the increased corpus cavernosum relaxation induced by SNP and EFS in the eNOS$^{-/-}$-HU group, indicating that eNOS deficiency impairs the action of hydroxyurea in the corpus cavernosum.

Among the limitations of this study, the lack of dose titration stands out. The hydroxyurea dosage of 100 mg/kg/day was chosen based on previous studies performed in mice that showed the efficacy of this dosage [26, 35]. However, this dosage is substantially higher than the 20–30 mg/kg/day doses commonly used in the treatment of adult patients with SCD [36]. Another limitation is that longer treatment periods were not evaluated. However, previous studies suggest that a treatment period of 2–3 weeks in mice is sufficient to induce significant changes in erectile function, particularly in the modulation of the NO-sGC-cGMP pathway [9, 10, 37].

## Conclusion

This study shows the limitations of hydroxyurea as a treatment for addressing priapism in mouse models of SCD and eNOS$^{-/-}$. Our findings indicate that NO generated from hydroxyurea may be neutralized by elevated plasma hemoglobin and ROS before it can effectively bind to GCs in the corpus cavernosum of SCD mice, thus preventing the normalization of NO/cGMP signaling (Fig 4). Furthermore, the downregulation of eNOS in the penis of SCD mice, which has already been reported, may impair the pharmacological action of hydroxyurea at the endothelial level. Given these limitations, our findings emphasize the urgency for exploring

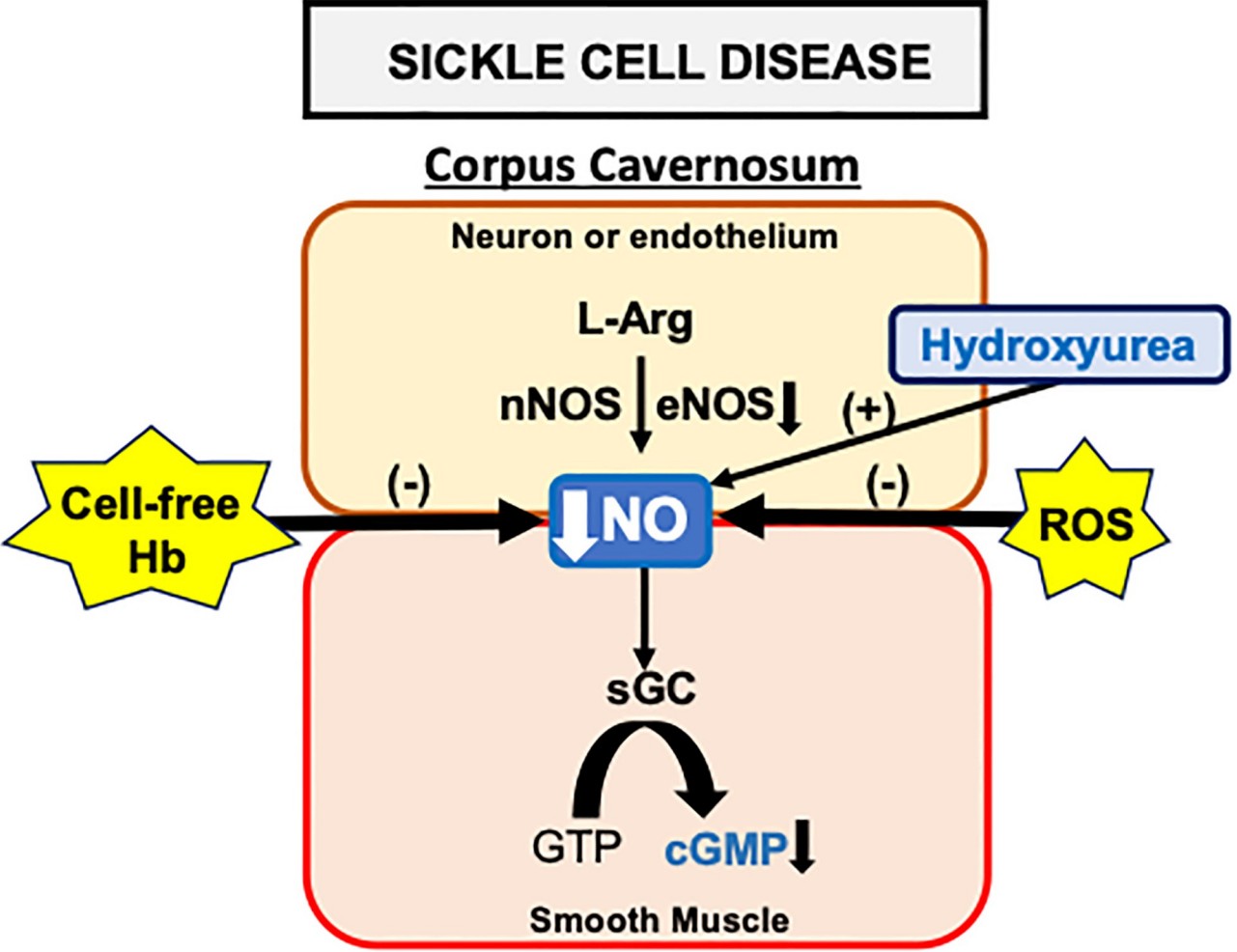

**Fig 4. The limited effectiveness of hydroxyurea is caused by the excess of cell-free hemoglobin and elevated concentrations of reactive oxygen species (ROS) in the corpus cavernosum of SCD mice, which neutralize the NO generated by the drug.** L-arg, L-arginine.

alternative therapeutic avenues for priapism in SCD that are not hindered by high plasma hemoglobin and ROS levels.

## Supporting information

**S1 Dataset.**
(XLSX)

## Author Contributions

**Conceptualization:** Pamela da Silva Pereira, Leonardo Oliveira Reis, Fernando Ferreira Costa, Fábio Henrique Silva.

**Data curation:** Danillo Andrade Pereira, Pamela da Silva Pereira, Tammyris Helena Rebecchi Silveira, Fabiano Beraldi Calmasini, Fernando Ferreira Costa, Fábio Henrique Silva.

**Formal analysis:** Danillo Andrade Pereira, Pamela da Silva Pereira, Tammyris Helena Rebecchi Silveira, Fabiano Beraldi Calmasini, Leonardo Oliveira Reis, Fernando Ferreira Costa, Fábio Henrique Silva.

**Funding acquisition:** Fernando Ferreira Costa, Fábio Henrique Silva.

**Investigation:** Dalila Andrade Pereira, Danillo Andrade Pereira, Pamela da Silva Pereira, Tammyris Helena Rebecchi Silveira, Fabiano Beraldi Calmasini, Fábio Henrique Silva.

**Methodology:** Dalila Andrade Pereira, Danillo Andrade Pereira, Pamela da Silva Pereira, Tammyris Helena Rebecchi Silveira, Fábio Henrique Silva.

**Project administration:** Fábio Henrique Silva.

**Writing – original draft:** Dalila Andrade Pereira, Danillo Andrade Pereira, Fábio Henrique Silva.

**Writing – review & editing:** Dalila Andrade Pereira, Danillo Andrade Pereira, Fábio Henrique Silva.

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
