## [Decision Letter · Decision Letter 0]

21 Aug 2023

PONE-D-23-17393Hydroxyurea does not reverse functional alterations of the nitric oxide-cGMP pathway associated with priapism phenotype in corpus cavernosum from sickle cell mousePLOS ONE

Dear Dr. Silva,

Thank you for submitting your manuscript to PLOS ONE. After careful consideration, we feel that it has merit but does not fully meet PLOS ONE’s publication criteria as it currently stands. Therefore, we invite you to submit a revised version of the manuscript that addresses the points raised during the review process.

Please consider reviewers' comments. 

We look forward to receiving your revised manuscript.

Kind regards,

Wesley Ribeiro, Ph.D.

Academic Editor

PLOS ONE

Reviewers' comments:

Reviewer's Responses to Questions

**Comments to the Author**

1. Is the manuscript technically sound, and do the data support the conclusions?

Reviewer #1: Partly

Reviewer #2: Yes

2. Has the statistical analysis been performed appropriately and rigorously? 

Reviewer #1: Yes

Reviewer #2: Yes

3. Have the authors made all data underlying the findings in their manuscript fully available?

Reviewer #1: Yes

Reviewer #2: Yes

4. Is the manuscript presented in an intelligible fashion and written in standard English?

Reviewer #1: Yes

Reviewer #2: Yes

5. Review Comments to the Author

Reviewer #1: The MS PONE-D-23-17393 investigated the effects of HU in SCD mouse model.

The purpose and results of the study are understandable.

The Dose resp and administration period ( 3 weeks ) of HU as a mouse model should be ideally checked and strengthened ( they mention it for future studies).

In fact, the Conclusion section, which should state a general conclusion, contains such sentences for exp conditions ( 3 weeks of treatments...).

As data set ( Fig.1-3 ) are rather compact, a summary illustration should be included as Fig 4 to appeal for wide range of readers other than erectile biology ;

( possibility of excess plasma hemoglobin and reactive oxygen species in SCD may react with NO related with alteration the nitric oxide-cGMP path)

Reviewer #2: This work was nicely conceived and conducted. The work seems rigorous, and the conclusion seem appropriate for the findings made. However, a major concern is whether the hydroxyurea dosing and treatment schedule was effective. How was the dosing scheduled determined? Is the dosing equivalent to pharmacologic dosing used of human level? Was a positive control of some sort or other manner confirming dosing used? It seems your final paragraph briefly stated a limitation with dosing. But this is not clear, and further discussion on this point seems appropriate.

6. PLOS authors have the option to publish the peer review history of their article (what does this mean?). If published, this will include your full peer review and any attached files.

Reviewer #1: No

Reviewer #2: No

---

## [Author Response · Author response to Decision Letter 0]

11 Sep 2023

Reviewer #1: The MS PONE-D-23-17393 investigated the effects of HU in SCD mouse model. The purpose and results of the study are understandable. 

The Dose resp and administration period ( 3 weeks ) of HU as a mouse model should be ideally checked and strengthened (they mention it for future studies). In fact, the Conclusion section, which should state a general conclusion, contains such sentences for exp conditions ( 3 weeks of treatments...). 

As data set ( Fig.1-3 ) are rather compact, a summary illustration should be included as Fig 4 to appeal for wide range of readers other than erectile biology ; ( possibility of excess plasma hemoglobin and reactive oxygen species in SCD may react with NO related with alteration the nitric oxide-cGMP path).

Answer: First of all, we are grateful to this reviewer for his/her careful reading and thorough analysis of this manuscript, which certainly led to a much-improved manuscript. 

Regarding the hydroxyurea dosage, we chose a dose of 100 mg/kg, based on previous studies that evaluated this same dosage. One of these studies compared two different dosages (10 mg/kg and 100 mg/kg) and validated the efficacy of the higher dosage. (Meiler et al., 2011, Blood, doi: 10.1182/blood-2010-11-319137). In addition, other studies corroborate the effectiveness of this dosage in other contexts. (Haijer F, et al.. FASEB J. 2023 doi: 10.1096/fj.202300920R; Lebensburger et al., 2010, Haematologica, doi: 10.3324/haematol.2010.023325). As for the treatment time scheme, we adopted a period of 2-3 weeks, based both on our own experience and on the existing literature. (Silva FH et al., 2014, JSM, DOI: 10.1111/jsm.12682, Silva et al., JPET, 2016, doi: 10.1124/jpet.116.235473, Pinheiro et al., Plos One doi: 10.1371/journal.pone.0269310). Previous studies suggest that a treatment period of 2-3 weeks in mice is sufficient to induce significant changes in erectile function, particularly in the modulation of the NO-sGC-cGMP pathway (Musicki B, et al., JCP. 2021 doi: 10.1002/jcp.30075, de Oliveira MG et al., Andrology. 2023, doi: 10.1111/andr.13340; Bivalacqua et al., PLoS One. 2013 doi: 10.1371/journal.pone.0068028).

We rewrote a paragraph about the study limitation. Please see the discussion section, last paragraph, page 11.

We appreciate the suggestion to modify the conclusion. We take into account and improve the conclusion. Please see the new conclusion on page 11.

We appreciate the suggestion to insert an illustration that certainly improved our manuscript. We present the illustration in figure 4.

Reviewer #2: This work was nicely conceived and conducted. The work seems rigorous, and the conclusion seem appropriate for the findings made. However, a major concern is whether the hydroxyurea dosing and treatment schedule was effective. How was the dosing scheduled determined? Is the dosing equivalent to pharmacologic dosing used of human level? Was a positive control of some sort or other manner confirming dosing used? It seems your final paragraph briefly stated a limitation with dosing. But this is not clear, and further discussion on this point seems appropriate.

Answer: We thank the reviewer for constructive feedback and pertinent questions about our manuscript. The suggestions presented were very important to improve our manuscript. 

Regarding the hydroxyurea dosage, we chose a dose of 100 mg/kg, based on previous studies that evaluated this same dosage. One of these studies compared two different dosages (10 mg/kg and 100 mg/kg) and validated the efficacy of the higher dosage. (Meiler et al., 2011, Blood, doi: 10.1182/blood-2010-11-319137). In addition, other studies corroborate the effectiveness of this dosage in other contexts. (Haijer F, et al.. FASEB J. 2023 doi: 10.1096/fj.202300920R; Lebensburger et al., 2010, Haematologica, doi: 10.3324/haematol.2010.023325). As for the treatment time scheme, we adopted a period of 2-3 weeks, based both on our own experience and on the existing literature. (Silva FH et al., 2014, JSM, DOI: 10.1111/jsm.12682, Silva et al., JPET, 2016, doi: 10.1124/jpet.116.235473, Pinheiro et al., Plos One doi: 10.1371/journal.pone.0269310). Previous studies suggest that a treatment period of 2-3 weeks in mice is sufficient to induce significant changes in erectile function, particularly in the modulation of the NO-sGC-cGMP pathway (Musicki B, et al., JCP. 2021 doi: 10.1002/jcp.30075, de Oliveira MG et al., Andrology. 2023, doi: 10.1111/andr.13340; Bivalacqua et al., PLoS One. 2013 doi: 10.1371/journal.pone.0068028).

To provide clinical context, adult patients with sickle cell anemia are often treated with hydroxyurea doses ranging from 20-30 mg/kg/day.

We are grateful for your valuable suggestion to enhance the section discussing the study's limitations. Please see the discussion section, last paragraph, page 11.

In response to your question regarding the use of a positive control to validate the dosage, we would like to inform you that the same lot of hydroxyurea was used in previous studies performed in our laboratory, serving as a positive control. In these studies, hydroxyurea was shown to be effective in significantly increasing gamma globin levels in K562 cells. Although the experimental context was different, the demonstrated efficacy gives us confidence that the hydroxyurea used had its pharmacological activity preserved.

---

## [Editor Report · Decision Letter 1]

26 Sep 2023

Hydroxyurea does not reverse functional alterations of the nitric oxide-cGMP pathway associated with priapism phenotype in corpus cavernosum from sickle cell mouse

PONE-D-23-17393R1

Dear Dr. Silva,

We’re pleased to inform you that your manuscript has been judged scientifically suitable for publication and will be formally accepted for publication once it meets all outstanding technical requirements.

Kind regards,

Wesley L. C. Ribeiro, Ph.D.

Academic Editor

PLOS ONE
---

## [Editor Report · Acceptance letter]

29 Sep 2023

PONE-D-23-17393R1 

Hydroxyurea does not reverse functional alterations of the nitric oxide-cGMP pathway associated with priapism phenotype in corpus cavernosum from sickle cell mouse 

Dear Dr. Silva:

I'm pleased to inform you that your manuscript has been deemed suitable for publication in PLOS ONE. Congratulations! Your manuscript is now with our production department. 

Kind regards, 

on behalf of

Dr. Wesley Lyeverton Correia Ribeiro 

Academic Editor

PLOS ONE